# The Grouping Patterns of *Cervus canadensis songaricus* in the Tianchi Bogda Peak Nature Reserve of Tianshan Mountain, Northwestern China

**DOI:** 10.3390/ani15020247

**Published:** 2025-01-17

**Authors:** Xuejun Ma, David Blank, Feng Xu

**Affiliations:** 1Xinjiang Key Laboratory for Ecological Adaptation and Evolution of Extreme Environment Biology, College of Life Sciences, Xinjiang Agricultural University, Urumqi 830052, China; maxuejun9@outlook.com; 2State Key Laboratory of Desert and Oasis Ecology, Key Laboratory of Ecological Safety and Sustainable Development in Arid Lands, Xinjiang Institute of Ecology and Geography, Chinese Academy of Sciences, Urumqi 830011, China; 3Xinjiang Key Laboratory of Biodiversity Conservation and Application in Arid Lands, Xinjiang Institute of Ecology and Geography, Chinese Academy of Sciences, Urumqi 830011, China; 4Sino-Tajikistan Joint Laboratory for Conservation and Utilization of Biological Resources, Urumqi 830011, China; 5CAS Research Center for Ecology and Environment of Central Asia, Bishkek 720001, Kyrgyzstan; blankdavid958@yahoo.com; 6University of Chinese Academy of Sciences, Beijing 100049, China

**Keywords:** grouping patterns, group size, group types frequency

## Abstract

The grouping patterns of the *Cervus canadensis songaricus* inhabiting the Bogda Peak Nature Reserve, Tianshan Mountains, Northwest China, were investigated from July 2019 to November 2020 using infrared-triggered camera-traps. Group size and composition were analyzed, revealing a range of 1 to 32 individuals per group, with the majority of groups consisting of 1 to 9 individuals. Both group size and frequency of different group types exhibited significant seasonal variation. These fluctuations were closedly associated with reproductive cycles and seasonal changes, with notable sex-specific aggregations observed in July and from October to December. The results suggested that Tianshan wapiti adjust their group sizes and compositions in response to environmental conditions and reproductive demands. These findings provide valuable insights into the behavioral ecology of Tianshan wapiti and have important implications for their conservation and management in China.

## 1. Introduction

Understanding grouping patterns is pivotal for assessing the lifestyles of mammals, particularly gregarious ungulates. Analyzing the grouping patterns of various ungulate species allows for a deeper exploration of their social organization, with group size and composition serving as fundamental components of this dynamic [1,2,3]. Variability exists not only across different ungulate species, where group characteristics can differ significantly, but also within the same species [4]. The prevailing explanation for the high level of aggregation observed in ungulates is the reduction of predation risk, achieved through enhanced predator detection (the “many eyes” effect) and dilution effects [5,6].

However, many herbivores form small groups due to habitat constraints that may not facilitate efficient predator detection. Consequently, factors beyond predation pressure, such as food availability and quality, reproductive cycles, population density, and distance at which individuals can perceive one another, significantly influence group characteristics [7,8]. Ungulates must balance the benefits of social living by minimizing predation risk while also reducing competition for resources [9]. This necessitates a trade-off between forming larger groups and ensuring individual fitness [10]. Moreover, in many mammalian species, males and females often live solitarily or in separate groups outside the breeding season [11]. Sexual segregation is particularly prevalent among ungulates [12], although its underlying causes remain debated. These causes may depend on species-specific factors and involve sex-related differences in food preferences, habitat selection, energy expenditure, or activity rhythm [13].

In this study, we examined the grouping patterns of *Cervus canadensis songaricus* [14,15,16,17,18,19,20,21,22,23,24] inhabiting the Tianchi Bogda Peak, Xinjiang, China. The Tianshan wapiti is primarily distributed in the alpine zone of mixed coniferous and broad-leaved forests, forest shrubs, and grasslands on the northern slopes of the Tianshan Mountains at elevations ranging from 1800 to 3200 m. We analyzed the seasonal variations in group size among Tianshan wapiti, as seasonality is expected to influence group dynamics. Additionally, we tested whether the observed groups throughout the annual cycle resulted from sex-independent associations by applying the sexual segregation and aggregation statistic (SSAS) to the numbers of adult males and females within the groups.

## 2. Materials and Methods

### 2.1. Study Area

The study was conducted in the Tianchi Bogda Peak Nature Reserve, Xinjiang, China (43°45′–43°59′ N, 88°00′–88°20′ E). This reserve is situated on the eastern slopes of Bogda Peak in the mid-latitude region of Asia, far from oceanic influences. The total area of the reserve encompasses 38,069 km^2^ [25], featuring a diverse array of natural landscapes, including snow-capped mountains, glaciers, lakes, forests, and meadows. Elevations within the reserve range between 1300 to 5445 m above sea level. The annual average precipitation is between 600 and 700 mm, primarily occurring from April to September, while the annual average evaporation is approximately 1439 mm. The climate is characterized as temperate continental, with long, cold winters and short, hot summers. The annual average temperature is around 2 °C, with the hottest month (July) averaging 15.9 °C and the coldest month (January) averaging −12.4 °C [25]. Notably, the average annual temperature remains −9 °C above the snow line. The average frost-free period is 98.4 days, and relative humidity ranges from 70% to 85%. The predominant vegetation types include alpine cushion vegetation, alpine meadow, sub-alpine meadow, mountain evergreen coniferous forest, and meadow steppe [25]. The reserve is home to 184 species of wild vertebrates across 56 families and 25 orders, with 127 species listed on the IUCN Red List, including Eurasian lynx (*Lynx lynx*), Asian badger (*Meles leucurus*), and snow leopard (*Panthera uncia*) [25,26].

### 2.2. Data Collection

The monitoring period for this study spanned July 2019 to November 2020. Based on the climatic characteristics of the reserve, the year was divided into four seasons: spring (April to May), summer (June to September), autumn (October to November), and winter (December to March) [27]. Unlike previous studies that employed line transect census methods, this research utilized infrared surveillance cameras to monitor various locations within the protected area. We established 30 infrared camera sites across different habitats, including forested habitats (1500~2600 m), alpine meadows, and bare rock habitats above the forest line (2600~3800 m). Each site was spaced at least 300 m apart and was not baited. Sites selection was based on the presence of animal activity indicators (e.g., animal trails, feces, and food remains), field of view, terrain flatness, sun exposure angle, and shading conditions provided by surrounding vegetation. A total of 58 infrared cameras were deployed at these sites. All photographs of wapiti captured by the infrared cameras were viewed individually, and detailed records were maintained for each valid photograph, including the date, time, temperature, and location. Individual wapiti were identified based on distinctive characteristics such as coat color, body size, sex, and age classes.

While analyzing the images from the surveillance cameras, we distinguished the sex and age of wapiti using the following criteria:(1)Adult males: large-sized normally with branched antlers; when antlers cast in late winter-early spring, there are visible signs of shedding on the head.(2)Adult females: medium-sized, antlerless and often accompanied by calves.(3)Yearlings: smaller than adults and, if males, provided with non-branched and slender antlers.

Seven group types were distinguished:(1)Single female groups: only one adult female.(2)Single male groups: only one adult male.(3)Female groups: groups consisting of two or more adult females.(4)Male groups: groups consisting of two or more adult males.(5)Mixed groups: groups consisting of females and males.(6)Mother-kid groups: groups consisting of females and kids, excluding males.(7)Young groups: containing subadults or calves.

### 2.3. Data Analysis

As the distribution of group sizes deviated significantly from normality, seasonal differences in mean group size were assessed using the Kruskal-Wallis H-test. Seasonal variations in the observation frequencies of the different group types were analyzed using R × C χ^2^-test. In addition, we tested whether the groups observed throughout the annual cycle could be the result of sex-independent associations, applying the sexual segregation and aggregation statistic (SSAS) to the numbers of adult males and females in the groups [28]. All statistical analyses were performed using SPSS 26.0 and R 4.2.1.

## 3. Results

Throughout the study period, we observed a total of 609 groups, whose 2335 members were 601 adult males (i.e., 25.74% of all individuals), 1529 adult females (65.48%), and 205 young individuals (8.78%).

### 3.1. Overall Group Size Distribution

Throughout the study period, wapiti’s groups varied by size from 1 to 32 individuals (*n* = 609). However, the frequency of group size had an obvious decline trend with increase of group size value. The single wapiti individuals constituted 31.03% of all recorded groups, the groups of two made up 21.84%, the proportion of groups of three individuals composed 12.48%, and groups of size less than or equal to 10 accounted for 92.78%, with only 7.22% of wapiti individuals existing in groups larger than 10 individuals in size (Figure 1). The skewness of the distribution was large when the number of individuals seen in groups of different sizes was considered. The smallest groups were the most numerous and fluctuated mainly between 1 and 6 individuals, accounting for 84.90% of groups and 56.88% in terms of individuals. The smallest groups were comprised of 7 and 9 individuals (7.39% and 15.12% respectively).

### 3.2. Seasonal Variability in Group Size

Kruskal-Wallis H-test revealed significant seasonal differences in group size (*H* = 28.91, *df* = 3, *p* < 0.0001) with a maximum group size in spring (Figure 2; spring vs. any other season: spring vs. summer: *H* = 19.69, *df* = 1, *p* < 0.01; spring vs. autumn: *H* = 24.25, *df* = 1, *p* < 0.01; spring vs. winter: *H* = 18.525, *df* = 1, *p* < 0.01;). Similar results were obtained when group size was calculated ignoring calves, showing that the spring maximum was not due to newborns (Figure 2; four seasons: *H* = 22.59, *df* = 3, *p* < 0.0001; spring vs. any other season: spring vs. summer: *H* = 17.85, *df* = 1, *p* < 0.01; spring vs. autumn: *H* = 19.13, *df* = 1, *p* < 0.01; spring vs. winter: *H* = 13.267, *df* = 1, *p* < 0.01).

Furthermore, we also found seasonal differences in group size for mixed groups (*H* = 12.40, *df* = 3, *p* < 0.01) and mother-kid groups (*H* = 8.32, *df* = 2, *p* < 0.02). Size of mixed groups was significantly larger in spring than in summer (*H* = 31.99, *df* = 1, *p* < 0.0001) and autumn (*H* = 40.00, *df* = 1, *p* < 0.0001), and size of mother-kid groups were much larger in spring than summer (*H* = 23.24, *df* = 1, *p* < 0.0001; Table 1; Figure 3).

### 3.3. Seasonal Variability in Frequency of Different Group Types

The 609 groups observed throughout the study period were 91 single adult females (i.e., 14.94% of all groups), 92 single adult males (15.11%), 121 female groups (19.87%), 54 male groups (8.87%), 196 mixed groups (32.18%), 48 mother-kid groups (7.89%), and seven young groups (1.15%). Nonetheless, group type frequencies varied according to season (χ^2^ = 93.32, df = 18, *p* < 0.05; Figure 4).

In spring, mixed groups were the most frequent group type, while single adult females, single adult males, female groups, and male groups exhibited their lowest frequencies of the year. In summer, the frequency of mixed groups fell to its lowest value, while that of all other group types increased. In autumn, the frequency of single adult males reached its highest value while that of male groups was back to its lowest value. Simultaneously, the frequency of mixed groups increased again, and female-young groups were no longer observed. In winter, frequencies of single adult males and mixed groups decreased, while those of the other group types re-increased.

### 3.4. Sex Segregation

While analyzing the Tianshan wapiti each month using SSAS, we observed no segregation of female and male Tianshan wapiti in any of the months. However, sex aggregation was found in July and from October to December (Figure 5).

## 4. Discussion

The group size of ungulates is influenced by several factors, with habitat openness, resource abundance and availability, and predator pressure being the most significant [29]. Other potential influences were not considered in this study. It is well established that ungulates tend to form larger groups in open environments, a conclusion supported by extensive research [1]. Generally, larger animal groups demand more resources, leading to increased competition for survival among individuals. Conversely, individuals in larger groups are assumed to experience reduced predator pressure. Our research in the Bogda Peak Nature Reserve indicates that most wapiti groups are small, typically comprising 1 to 6 individuals, with a pronounced right-skewed distribution (Figure 1). The number of observed groups decreases as group size increases. Wapiti inhabit the rugged and steep areas of the reserve, characterized by seasonal climate changes and uneven resource distribution, resulting in an unstable food supply. Consequently, wapiti tend to form smaller groups to minimize intraspecific competition for food. Similar behavior is observed in Siberian ibex in the Eastern Tien-Shan Mountains, which form small groups to reduce foraging competition, adapting to seasonal changes in local vegetation conditions [6]. Additionally, based on our camera-trap survey of the birds and animals in the Bogda Peak study area [26], we found that Tianshan wapiti has very few predators; wolves (*Canis lupus*) were never observed, and Eurasian lynx only rarely detected. Therefore, Tianshan Wapiti do not need to form large groups to mitigate predation risk.

The reproductive cycle of Tianshan wapiti significantly influences the frequency of group formation [30]. These wapiti rut and mate from September to October, with calves born between May and July. During the summer months (June to September), when female Tianshan wapiti separate from the group to give birth and nurture their calves, the number of mixed-sex groups decreases, while the number of female-only groups increases. Like other ungulates, male wapiti typically display reproduction-related behaviors only during the mating season, leading to an increase in male and single-male groups during this period [31,32]. The frequency of occurrence of single adult males increased as Tianshan wapiti entered the autumn (October-November) to reduce inter-autumn male competition and increase the mating success of adult males. We found that birth increased mean group size by 1 to 2 individuals by comparing group changes between all individuals (n = 2335) and only adults (n = 2130) but did not result in a significant difference (Figure 2). The mother-kid groups in this study area accounted for 7.88% of the total group size of Tianshan wapiti, while the female group accounted for 19.87% of the total group size, which was much higher than that of the mother-kid groups. This indicates that the proportion of survival and the proportion of the number of surviving calves in the study area is low for female equids, resulting in a low frequency of occurrence of mother-kid groups. Mixed groups were large and had a high frequency of occurrence (Figure 2 and Figure 4).

According to our results, Tianshan wapiti gather in larger herds during spring than at any other time of the year. Correlatively, spring groupings frequently include adults of both sexes. In the same season, mother-calf groups are much rarer than mixed herds, although exceptionally large during the calving period. In summer, group sizes decrease significantly compared to spring. Consequently, mixed groups become much rarer, while all group types (including solitary adults) become more common than in the previous season. In autumn, while group sizes remain limited, the rut leads to a decrease in male groups, an increase in the frequency of single adult males, and a resurgence of mixed groups, while female groups are no longer observed. In winter, following the rutting season, mean group size increases only slightly, and all group types are re-observed, albeit with a low frequency of single adult males.

Given that Tianshan wapiti exhibit strong sexual dimorphism in body size, one might expect adult males and females to primarily form single-sex groups [33,34,35,36]. However, we found no evidence of sexual segregation at any time of the year; rather, the two sexes tended to aggregate during July and from October to December. There was no sex segregation between female and male wapiti in any of the other months. This finding contrasts with Bonenfant’s [28] study of European red deer, which reported significant sexual segregation from April to June, along with significant sexual aggregation in November, January, and February. The difference may arise from the fact that red deer do not gather in large herds in April and May as Tianshan wapiti do, which could explain the observed discrepancies in spring.

This difference may be related to food resources and food quality within the reserve. The vegetation is primarily composed of a mountainous cold-temperate coniferous forest belt dominated by Picea schrenkiana [37], with limited mixing of other tree species. Snowy ridge spruce constitutes the main winter food source for Tianshan wapiti (28%), followed by grasses (Poaceae), sedges (Cyperaceae), and mosses (Bryophyta) [38]. We hypothesize that the decrease in average group size between spring and summer is primarily due to changes in forage distribution and the food requirements of wapiti post-winter. The activity patterns of ungulates are driven by fluctuations in the quantity and quality of available food. During the summer months, Tianshan wapiti have access to abundant, easily digestible food resources, reducing the need for competition. Consequently, their movement time spent foraging decreases, allowing for more resting time. This results in a smaller distance between nearest neighbors and the observed phenomenon of sex aggregation in July. During the autumn rut, male and female wapiti congregate. However, Tianshan wapiti face reduced food resources during winter months, with snowfall further complicating access to food. The scarcity of food during this season increases survival challenges and energy expenditure for wapiti [39]. According to the “rumen fill theory,” they adapt their activities in response to changes in food availability [33]. Consequently, during winter, wapiti prefer to rest under the canopies of dense vegetation and in flat depressions to minimize snow depth, reduce movement costs, and maintain body warmth. However, the limited availability of such habitats leads to the congregation of wapiti.

## 5. Conclusions

In summary, the group size and frequency of Tianshan wapiti are significantly influenced by the breeding cycle and seasonal changes. Calving during spring and summer increases the average group size by 1 to 2 individuals. Summer coincides with calving period for Tianshan wapiti, during which the frequency of mixed groups decreases while the occurrence of mother-calf and female-only groups increases. As autumn progresses, the average group size diminishes, accompanied by an increase in the frequency of mixed herds and solitary males. Notably, Tianshan Wapiti experience a low incidence of natural predators, with sparse populations of potential threats such as lynxes. Additionally, sex aggregation is observed in July, as well as from October to December.

## Figures and Tables

**Figure 1 animals-15-00247-f001:**
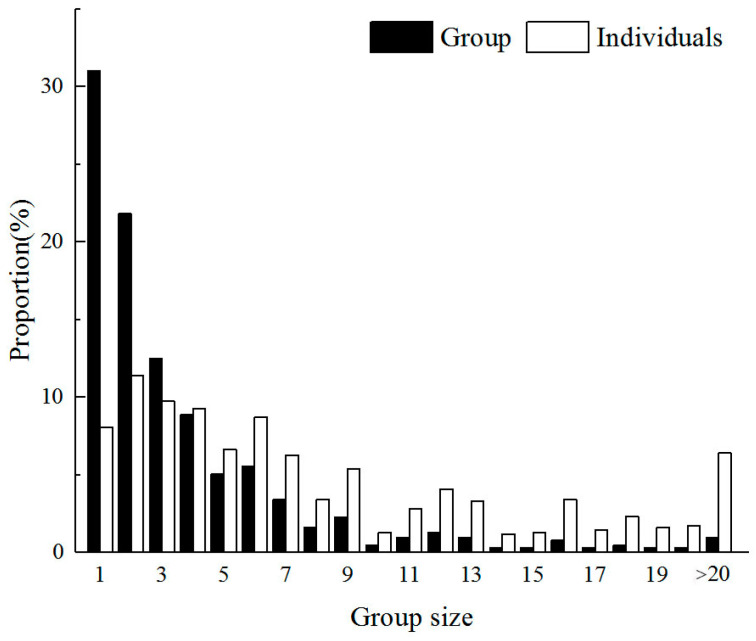
Group size of wapiti in the Tianchi Bogda Peak Nature Reserve: proportions of groups (*n* = 609) and individuals (*n* = 2335).

**Figure 2 animals-15-00247-f002:**
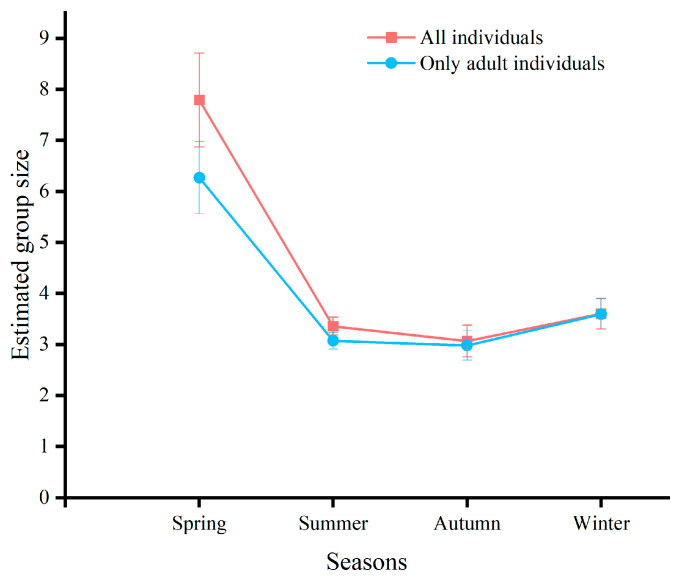
Seasonal variation in mean group change for all individuals (*n* = 2335) and mean group size for only adult individuals (*n* = 2130).

**Figure 3 animals-15-00247-f003:**
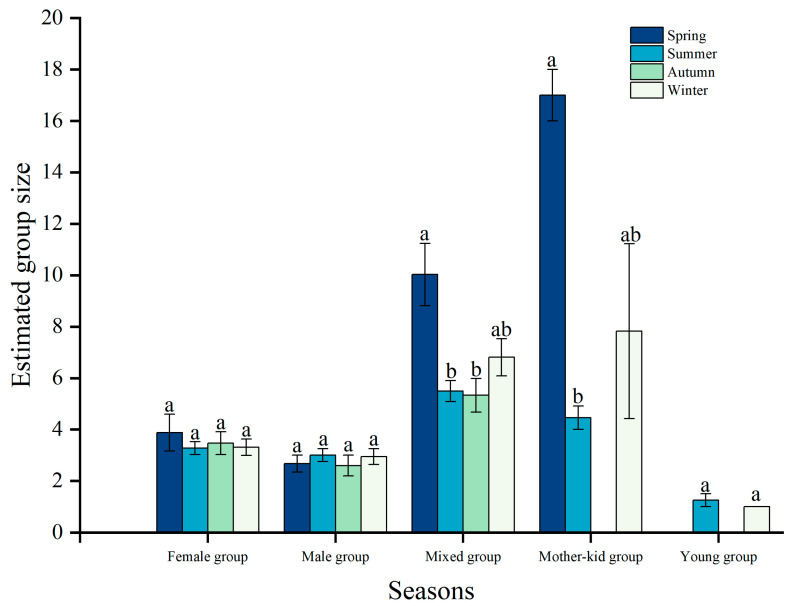
Seasonal changes in group size of different group types in Tianshan wapiti. Note: The letters (e.g., “a” and “b”) above the bars in the bar chart indicate statistically significant differences between the groups. Bars labeled with the same letter (e.g., ‘a’) are not significantly different from each other (*p* > 0.05). Bars labeled with different letters (e.g., ‘a’ vs. ‘b’) are significantly different (*p* < 0.05).

**Figure 4 animals-15-00247-f004:**
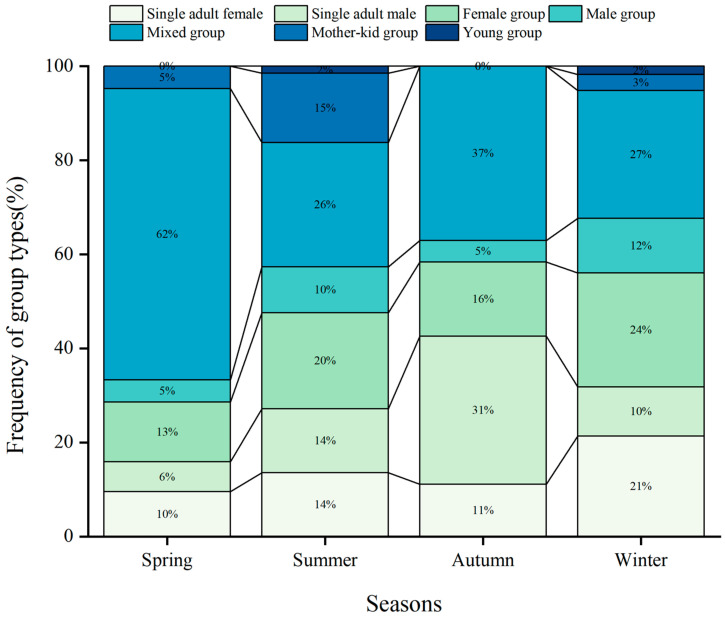
Seasonal variation in the frequency of occurrence of group types.

**Figure 5 animals-15-00247-f005:**
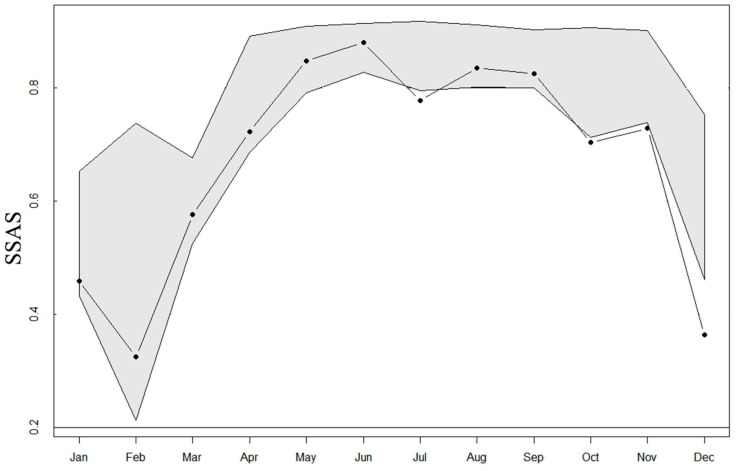
Monthly patterns of sexual segregation and aggregation between females and males in Tianshan wapiti. The SSAS indicates significant sexual segregation or aggregation when the observed value (black point) falls above or below the SSAS expected interval (grey area), respectively.

**Table 1 animals-15-00247-t001:** The mean value (±standard error) of different group types of Tianshan wapiti in different seasons in Tianchi Bogda Peak Nature Reserve.

Group Type	Season			
	Spring	Summer	Autumn	Winter
Single adult female	1	1	1	1
Single adult male	1	1	1	1
Female group	3.88 ± 0.72	3.28 ± 0.25	3.47 ± 0.44	3.31 ± 0.32
Male group	2.67 ± 0.33	3 ± 0.25	2.6 ± 0.40	2.95 ± 0.31
Mixed group	10.03 ± 1.21	5.49 ± 0.41	5.33 ± 0.65	6.81 ± 0.72
Mother-kid group	17 ± 1	4.46 ± 0.45	—	7.83 ± 3.4
Young group	—	1.25 ± 0.25	—	1
All individuals	7.79 ± 0.92	3.36 ± 0.17	3.06 ± 0.31	3.60 ± 0.30
Only adult individuals	6.27 ± 0.71	3.07 ± 0.16	2.98 ± 0.29	3.49 ± 0.29

“—” means no wa.

## Data Availability

The data supporting the results of this study can be found in the manuscript.

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
