# Peer review of "The Grouping Patterns of Cervus canadensis songaricus in the Tianchi Bogda Peak Nature Reserve of Tianshan Mountain, Northwestern China"

_animals, 2025, doi:10.3390/ani15020247_

Round 1

Reviewer 1 Report

Comments and Suggestions for Authors

Premise

Some zoologists, including the authors of this ms, continue to consider the Cervus-elaphus-complex a monophyletic large group of subspecies occurring from Europe to N America, despite the fact that genetic studies have confirmed since the late Nineties (Polziehn & Strobeck 1998, Kuwayama & Ozawa 2000, Randi et al. 2001, Ludt et al. 2004, Lorenzini et al. 2005, Heckeberg 2020, Mackiewicz et al. 2022) the need to split the superspecies in two or better three distinct species (Western red deer C. elaphus of Europe and N Africa, Central Asian red deer C. hanglu of Turkestan, Tarim Basin and Kashmir including the Chinese yarkandensis, and wapiti C. canadensis of China, Siberia and N America including the subspecies wallichii, kansuensis, mcneilli, alashanicus, xanthopygus, songaricus, sibiricus and the American canadensis). Also in the IUCN Red List the old superspecies is now divided into three species (Lovari et al. 2018 for C. elaphus, Brook et al. 2017 for C. hanglu and Brook et al. 2018 for C. canadensis). Therefore the Cervid living in the Tianchi Bogda (or Bogeda?) Peak should be named Tianshan wapiti Cervus canadensis songaricus.

The manuscript requires profound changes. The taxonomy must be updated, the definitions of age groups and more in general the specific terminology must be re-formulated, the discussion of the results must be improved. Important works on red deer and wapiti biology, which could clarify most of the results, are ignored. The analysis of the results is simplistic and superficial and the explanations in the discussion of the results are naïve. The idea that males participate in raising kids is untenable, just as it is difficult to think that females seek out males in winter to mate. The argument of sexual segregation is not treated at all.

You should better answer at least to the following questions:

Why the sexual spatial and social segregation is so limited in this wapiti population? Why mixed groups are so common during the calving and lactation period? (Is this an artefact due to the misidentification of sexes when males cast their antlers and are partly confused with females or it is a real phenomenon?) Why mixed groups are frequent also in winter, when the rutting season is finished? The present explanations are wrong.

Notes on specific points:

Line 18: better: The group behavior of wapiti (Cervus canadensis)…was investigated…

Line 24: …that Tianshan wapiti adapt…

Lines 29-30: Please reformulate the sentence, given the taxonomic reassessment of red deer-wapiti complex.

35-38: Please eliminate here the references to the statistical methods used, which will be clarified in a specific paragraph.

46: if the grouping patterns is consistent with the reproductive cycle, why mixed groups are more frequent in winter and not in autumn when matings take place? Please clarify here and in the paragraph on Results.

67: You should mention here the social and spatial sex segregation in dimorphic ungulates, in which the differential diet preferences, habitat selection, energy expenditure, activity rhythms of the two sexes often produce a clear separation between sexes (cf R.T. Bowyer 2022 “Sexual segregation in Ungulates”, Johns Hopkins University Press).

68-75: Please re-formulate the sentences in light of the latest taxonomic studies. Tarim red deer is Cervus hanglu yarkandensis, Altai wapiti is C. canadensis sibiricus and Tianshan wapiti is C. c. songaricus.

78-81: If you summarise the social behavior of the local wapiti, please add some quotation for your statements.  Do you know the classical work by Clutton-Brock et al. 1982 “Red deer: behaviour and ecology of two sexes”? But there are also many paper on group size for the North American wapiti  or elk (Proffitt et al. 2012 etc)   Please correct the nomenclature: female red deer, calves and yearlings; subadults are actually almost fully grown males, physiologically but not socially mature, which mostly form groups with adult males.

94: Not clear. Please clarify the surface extension in sqkm

96: please give the % surface of the main habitats

101: 984???

105-105: Does the wolf occur in the Reserve? Please clarify.

113: surveillance cameras. Please give (slightly) more details.

114-120: this part with the definitions of age and sex classes should be reformulate:

(1)   Adult males : large-sized normally with branched antlers ; when antlers cast in late Winter – early Spring, sex is determined by body size and external sexual organs

(2)   Adult females : medium-sized, antlerless and often accompanied by calves

(3)   Yearlings, smaller than adults and, if males, provided with non-branched and slender antlers.

166 : What did you mean with “visible signs of shedding” ? Antlers just cast or in early growing stage?

122: “. All observed…”

130: MGS, i.e. the arithmetic…

137: Wapiti

183-186: Please move these sentences in the paragraph Materials and Methods

189-192: Please include only two decimals, as you did in the previous page

197-206: Idem, two decimals

211: Lacation period (sic). Please put “Spring” or correct it into “lactation”

246: Bogda instead of Bogeda?

256-257: From the fig. 2 we learn that the mixed groups are particularly high during Spring, while in the text you state that mixed groups “peak during the rutting season” in Autumn and that they “increase during the mating season”. Please clarify and better explain: why during Spring mixed groups are so frequent? Is this because new grasses are more abundant? If resources are so available in Spring, this could be a good reason for aggregations of grazing deer of both sexes (see also what you write at lines 273-274).

261: Wapiti and red deer are not territorial

263: calves

266: your explanation about the winter high frequency of mixed groups is quite unlikely. Do you really think that females search for mating males? Have you really observed frequent mating behaviours in Winter? In healthy populations the rutting season is limited to two-three weeks in Autumn. You should formulate a more likely hypothesis, probably connected with the distribution of food sources.

269: solitary females

274: Males do not participate to the offspring rearing!

275-276: Actually the grouping pattern is not linked only to the reproductive cycle but to the abundance, quality and distribution of trophic resources.

Author Response

Respond to Reviewer#1’ comments

Some zoologists, including the authors of this ms, continue to consider the Cervus-elaphus-complex a monophyletic large group of subspecies occurring from Europe to North America, despite the fact that genetic studies have confirmed since the late Nineties (Polziehn & Strobeck 1998, Kuwayama & Ozawa 2000, Randi et al. 2001, Ludt et al. 2004, Lorenzini et al. 2005, Heckeberg 2020, Mackiewicz et al. 2022) the need to split the superspecies in two or better three distinct species (Western red deer C. elaphus of Europe and N Africa, Central Asian red deer C. hanglu of Turkestan, Tarim Basin and Kashmir including the Chinese yarkandensis, and wapiti C. canadensis of China, Siberia and N America including the subspecies wallichii, kansuensis, mcneilli, alashanicus, xanthopygus, songaricus, sibiricus and the American canadensis). Also in the IUCN Red List the old superspecies is now divided into three species (Lovari et al. 2018 for C. elaphus, Brook et al. 2017 for C. hanglu and Brook et al. 2018 for C. canadensis). Therefore the Cervid living in the Tianchi Bogda (or Bogeda?) Peak should be named Tianshan wapiti Cervus canadensis songaricus.

We sincerely thank the reviewer for careful reading. As suggested by the reviewer,we have named the deer living at Tianchi Bogda Peak Tianshan wapiti ( Cervus canadensis songaricus).

The manuscript requires profound changes. The taxonomy must be updated, the definitions of age groups and more in general the specific terminology must be re-formulated, the discussion of the results must be improved. Important works on red deer and wapiti biology, which could clarify most of the results, are ignored. The analysis of the results is simplistic and superficial and the explanations in the discussion of the results are naïve. The idea that males participate in raising kids is untenable, just as it is difficult to think that females seek out males in winter to mate. The argument of sexual segregation is not treated at all.

You should better answer at least to the following questions:

Why the sexual spatial and social segregation is so limited in this wapiti population? Why mixed groups are so common during the calving and lactation period? (Is this an artefact due to the misidentification of sexes when males cast their antlers and are partly confused with females or it is a real phenomenon?) Why mixed groups are frequent also in winter, when the rutting season is finished? The present explanations are wrong.

We feel great thanks for your professional review work on our article. According to your nice suggestions, we have made extensive corrections to our previous draft, the detailed corrections are listed below.

  1. We have re-customized the definition of age groups with the following modifications:

While analyzing pictures from surveillance cameras, we distinguished red deer sex by the presence or absence of antlers and their age by body and antler sizes and shape of antlers and by the following characteristics: (1) Adult males: large-sized normally with branched antlers; when antlers cast in late Winter – early Spring, there are visible signs of shedding on the head. Sex is determined by body size and external sexual organs. (2) Adult females: medium-sized, antlerless, and often accompanied by calves. (3) Sub-adult individuals: Smaller individuals with slender and non-forked horns are male sub-adults.(4) Fawns: smaller, antlerless, usually following adult females; those born in the current year have a spotted pattern on both sides of the body.

  1. We have redefined the definition of the season.Based on the reserve's climatic characteristics, the year was divided into spring (April to May), summer (June to September), autumn (October to November), and winter (December to March)
  2. We have revised the Methods section to provide a more detailed description of the data analysis methods. Additionally, we have included a SASS analysis in the Results section.
  3. We have expanded the Discussion section to provide a more in-depth analysis of our research results. Specifically, we have compared our findings with existing studies. These additions aim to highlight the significance and contribution of our research.

Notes on specific points:

Line 18: better: The group behavior of wapiti (Cervus canadensis)…was investigated…

We sincerely thank the reviewer for careful reading. As suggested by the reviewer, we have corrected it to “The group behavior of wapiti (Cervus canadensis) living in the Bogeda Peak Nature Reserve, Tianshan Mountain, Northwest China, was investigated from July 2019 to November 2020.”

Line 24: …that Tianshan wapiti adapt…

As suggested by the reviewer, we have corrected it to “…that Tianshan wapiti adapt…”

Lines 29-30: Please reformulate the sentence, given the taxonomic reassessment of red deer-wapiti complex.

We sincerely thank the reviewer for careful reading. It has been reformulated as:”Being poorly known about the social organization of the subspecies of wapiti inhabiting the Tian Shan Mountains.”

35-38: Please eliminate here the references to the statistical methods used, which will be clarified in a specific paragraph.

We sincerely thank the reviewer for careful reading. It's been deleted.

46: if the grouping patterns is consistent with the reproductive cycle, why mixed groups are more frequent in winter and not in autumn when matings take place? Please clarify here and in the paragraph on Results.

We sincerely thank the reviewer for careful reading. We reclassified the seasons, and we know from the literature the breeding and rutting periods of the Tianshan wapiti, which helped us to analyze the grouping patterns.

Based on the reserve's climatic characteristics, the year was divided into spring (April to May), summer (June to September), autumn (October to November), and winter (December to March).Tianshan wapiti rut and mate in September-October, and give birth to fawns in May-July.

The highest frequency of male groups was observed in winter (11.56%) and the lowest frequency of male groups was observed during both the spring and autumn. The highest frequency of single male groups (31.48%) occurred during the autumn and the lowest frequency (9.52%) occurred during the spring.The frequency of female groups was highest in winter (24.27%) and lowest in the spring(12.69%), the frequency of female groups showed a decreasing trend from winter to spring, an increasing trend from spring to summer, and a decreasing trend from summer to autumn.The frequency of mixed group clustering also varied between seasons, with the highest frequency (61.90%) occurring during spring. Maternal clusters had the highest frequency (27.34%) in summer and 0 in the autumn. The Ewe-lamb group occurred four times in summer and three times in winter.

Fig. 4 Seasonal variation in the frequency of occurrence of group types

67: You should mention here the social and spatial sex segregation in dimorphic ungulates, in which the differential diet preferences, habitat selection, energy expenditure, activity rhythms of the two sexes often produce a clear separation between sexes (cf R.T. Bowyer 2022 “Sexual segregation in Ungulates”, Johns Hopkins University Press).

We sincerely appreciate the valuable comments. We have checked the literature carefully and added more discussion of social and spatial sex segregation in dimorphic ungulates to the INTRODUCTION section of the revised manuscript.”In many mammals, males and females live solitarily or in separate groups outside the breeding season . Sexual segregation is widespread in ungulates, and most ungulate species with hermaphroditic bodies live in separate groups outside of the breeding season. The underlying factors that cause sex segregation are poorly known. In this context, differences between the sexes in dietary preferences, habitat selection, energy expenditure, and activity rhythms of the two sexes often produce a clear separation between sexes.”

68-75: Please re-formulate the sentences in light of the latest taxonomic studies. Tarim red deer is Cervus hanglu yarkandensis, Altai wapiti is C. canadensis sibiricus and Tianshan wapiti is C. c. songaricus.

We have re-written this part according to the Reviewer's suggestion.”The taxonomy of red deer is complex; it includes 22 subspecies worldwide, with three subspecies identified in Xinjiang: Altai (Cervus canadensis sibiricus), Tianshan (C. c. songaricus), and Tarim (Cervus hanglu yarkandensis). “

78-81: If you summarise the social behavior of the local wapiti, please add some quotation for your statements. Do you know the classical work by Clutton-Brock et al. 1982 “Red deer: behaviour and ecology of two sexes”? But there are also many paper on group size for the North American wapiti or elk (Proffitt et al. 2012 etc) Please correct the nomenclature: female red deer, calves and yearlings; subadults are actually almost fully grown males, physiologically but not socially mature, which mostly form groups with adult males.

We sincerely appreciate the valuable comments. We scrutinized the literature, and since the social behavior of local wapiti was not documented, we chose to delete these sentences. The definition of age group was also corrected.

94: Not clear. Please clarify the surface extension in sqkm

We feel sorry for our carelessness. In our resubmitted manuscript, the typo is revised. Thanks for your correction.We have corrected the"hm2" into "km2".

96: please give the % surface of the main habitats

This study mainly utilized infrared camera technology to study the Bogda Peak Tianshan wapiti. The specific infrared camera deployment sites are shown in the following figure.

101: 984???

We feel sorry for our carelessness. In our resubmitted manuscript, the typo is revised. Thanks for your correction.We have corrected the"984" into "98.4".

105-105: Does the wolf occur in the Reserve? Please clarify.

According to our survey of bird and animal resources in the Bogda Peak study area, we found that the number of natural enemies of the Tianshan wapiti was sparse, wolves were not monitored by us, and the RAI of the lynx = 0.032 (relative abundance index) (Bai et al.2022).

113: surveillance cameras. Please give (slightly) more details.

We sincerely appreciate the valuable comments.We have added more details about Surveillance Cameras.”We selected 30 infrared camera sites in forested habitats (1,500 ~ 2,600 m) and areas of alpine meadow and bare rock habitats above the forest line (2,600 ~ 3,800 m). Each site was spaced at least 300 meters apart and was not baited. The sites were selected based on animal activity traces (e.g., animal paths, feces, foodstuffs), field of view, the flatness of the terrain, sun exposure angle, and shading conditions of the surrounding vegetation. A total of 58 infrared cameras were deployed at these sites.”

114-120: this part with the definitions of age and sex classes should be reformulate:

We sincerely appreciate the valuable comments. We have reformulated the definitions of age and sex class. “While analyzing pictures from surveillance cameras, we distinguished red deer sex by the presence or absence of antlers and their age by body and antler sizes and shape of antlers and by the following characteristics: (1) Adult males: large-sized normally with branched antlers; when antlers cast in late Winter – early Spring, there are visible signs of shedding on the head. Sex is determined by body size and external sexual organs. (2) Adult females: medium-sized, antlerless, and often accompanied by calves. (3) Sub-adult individuals: Smaller individuals with slender and non-forked horns are male sub-adults.(4) Fawns: smaller, antlerless, usually following adult females; those born in the current year have a spotted pattern on both sides of the body.”

166 : What did you mean with “visible signs of shedding” ? Antlers just cast or in early growing stage?

Antlers that are freshly shed and those that regrow after shedding will have visible signs of shedding, as shown below.

122: “. All observed…”

We sincerely thank the reviewer for careful reading. We agree with the reviewers' suggestions and revised the manuscript as suggested.

130: MGS, i.e. the arithmetic…

We sincerely thank the reviewer for careful reading. We agree with the reviewers' suggestions and revised the manuscript as suggested.

137: Wapiti

We sincerely thank the reviewer for careful reading. We agree with the reviewers' suggestions and revised the manuscript as suggested.

183-186: Please move these sentences in the paragraph Materials and Methods

We sincerely thank the reviewer for careful reading. We agree with the reviewers' suggestions and revised the manuscript as suggested.The sentences have been moved to the paragraph “Materials and methods”.

189-192: Please include only two decimals, as you did in the previous page

We sincerely thank the reviewer for careful reading. We agree with the reviewers' suggestions and revised the manuscript as suggested.

197-206: Idem, two decimals

We sincerely thank the reviewer for careful reading. We agree with the reviewers' suggestions and revised the manuscript as suggested.

211: Lacation period (sic). Please put “Spring” or correct it into “lactation”

We sincerely thank the reviewer for careful reading. We agree with the reviewers' suggestions and revised the manuscript as suggested.

246: Bogda instead of Bogeda?

Thanks for the suggestion. Yes it should be the Bogda. We revised it as suggested in the updated revisied version of the manuscript.

256-257: From the fig. 2 we learn that the mixed groups are particularly high during Spring, while in the text you state that mixed groups “peak during the rutting season” in Autumn and that they “increase during the mating season”. Please clarify and better explain: why during Spring mixed groups are so frequent? Is this because new grasses are more abundant? If resources are so available in Spring, this could be a good reason for aggregations of grazing deer of both sexes (see also what you write at lines 273-274).

We agree with the reviewers' suggestions and revised the manuscript as suggested.

261: Wapiti and red deer are not territorial

We sincerely thank the reviewer for careful reading. We agree with the reviewers' suggestions and revised the manuscript as suggested.

263: calves

We sincerely thank the reviewer for careful reading. We agree with the reviewers' suggestions and revised the manuscript as suggested.

266: your explanation about the winter high frequency of mixed groups is quite unlikely. Do you really think that females search for mating males? Have you really observed frequent mating behaviours in Winter? In healthy populations the rutting season is limited to two-three weeks in Autumn. You should formulate a more likely hypothesis, probably connected with the distribution of food sources

We agree with the reviewers' suggestions and revised the manuscript as suggested.We have rewritten the conclusion along with the rest of the manuscript.

269: solitary females

We sincerely thank the reviewer for careful reading. We have rewritten the conclusion along with the rest of the manuscript.

274: Males do not participate to the offspring rearing!

We sincerely thank the reviewer for careful reading. We agree with the reviewers' suggestions and revised the manuscript as suggested.

275-276: Actually the grouping pattern is not linked only to the reproductive cycle but to the abundance, quality and distribution of trophic resources.

We agree with the reviewers' suggestions and revised the manuscript as suggested.We have rewritten the conclusion along with the rest of the manuscript.

We are grateful for the invaluable feedback provided by the reviewers, and we look forward to the opportunity to improve our manuscript in response to their guidance.

Reviewer 2 Report

Comments and Suggestions for Authors

Please see attached pdf file.

Author Response

Respond to Reviewer#2’ comments

The manuscript submitted by Ma et al. investigates the grouping patterns in a population of the red deer subspecies inhabiting the Tianshan Mountains. The social organisation of this subspecies being poorly known, the data are undoubtedly of interest. In my opinion,however, the manuscript needs major revisions before it can be accepted for publication in Animals.

A first problem relates to the individual-centred measure that the authors use to characterize group size. To begin with, let me note that using such a measure (the ‘crowding index’) is not imposed by the manuscript’s subject: the authors study grouping patterns, not individual fitness or any other individual feature assumed to be correlated with group size. Second, using such an individual-centred measure is not particularly consistent with the rest of the manuscript since the authors also use group-centred statistics: they examine the frequencies of group types, not the frequencies with which individuals were found in the different group types. Finally, as Reiczigel et al (2008) point out, ‘crowding data’ are (by construction) non-independent, which makes it difficult to use standard statistical tools (whether parametric or not). The authors of the manuscript use a standard mixed effect model (GLMM) with group identity as random factor. The introduction of group identity as random factor in the model is certainly an interesting solution to the problem of data non-independence. However, while the model used does work on crowding data, the group size estimates it gives cannot be those expected, i.e. the group size in which the individual is found on average1. So, what is really gain by deriving (non-independent) crowding data from (independent) group size data, and then analysing them with this sophisticated model?

Clearly, in an article on grouping patterns, it would be simpler to investigate group size variations, using group size rather than the crowding index. Although group size data are far from normally distributed and subject to heteroscedasticity, their variations can be analysed using standard F-tests, provided P-values are recomputed by random permutation (e.g. of season labels within the table listing observed group sizes if differences between seasons are examined). Such a recalculation of P-values can easily be carried out using the R package lmPerm. Alternatively, and this is certainly the simplest solution, standard non-parametric tests (Kruskal-Wallis and Mann-Whitney tests) can be used instead of F and t tests.

Another point bothers me. The manuscript examines group composition and the size of the different group types. Surprisingly, however, the authors consider nowhere the inescapable links between observed sex ratio, group composition and group size. Nor do they investigate the related subject of sexual segregation.

Let’s start by assuming that population size is large, that adults have the same tendency to gather in groups and that they associate independently of their sex. Furthermore, let p be the proportion of females among adults in the population. In this case, the probability of being a female group for a group including n adults is pn, its probability of being a male group is (1–p)n, and its probability of being a mixed-sex group is 1 – pn – (1–p)n. Clearly, the probability of being a mixed-sex group increases with group size; as a consequence, the frequency of mixed-sex groups can be expected to increase as mean group size increases (for example, between seasons). According to the above formulas, the frequencies of female groups, male groups and mixed-sex groups should also vary when adult sex ratio varies (for example because individuals belonging to one of the two sexes tend to leave the sampled areas between two seasons). In addition, even if group size distribution (all group types combined) remains constant, variation in the sex ratio will entail variations in the average size of the different group types. This can be easily deduced from the above formulas and a fixed group size distribution, using (for each group type) the expected numbers of groups calculated for the different sizes observed.

These links between sex ratio, group composition and group size should be taken into account when interpreting the results obtained on the frequency and size of group types. Now, in several red deer subspecies (e.g. Conradt 1999; Peterson & Weckerly 2017), adults of the two sexes tend to make up separate groups. I do think that the authors should test for such a sexual segregation in the red deer subspecies of the Tianshan Mountains. There are several ways to do so. One way is to compare the observed frequencies of group types with those expected under the hypothesis of no sexual segregation, the latter frequencies being computed using the formulas given above and the group size distribution observed at each season (e.g. Cransac et al 1998). Another way is to use the segregation coefficient SC developed by Conradt (1998), or the sexual segregation and aggregation statistic SSAS proposed by Bonenfant et al (2007). Though I tend to prefer SC, testing sexual segregation using SSAS is easier since the authors provided a program to run on R with their article (an updated version of this program, obtained from the authors, is given at the end of the present report).

We sincerely thank the reviewer for their valuable feedback, which we have used to improve the quality of our manuscript. As you are concerned, several problems need to be addressed. According to your suggestions, we have made extensive corrections to our previous draft; the detailed corrections are listed below. Our response is given in blue text.

  1. We have selected “Mean ± Standard Error” for the measure describing the size of the group in accordance with your suggestion.
  2. Based on the time interval of the photos and the comparison of before and after photos, the sex ratio, adult and juvenile ratio, cluster size, and type of Tianshan wapiti were counted. Red deer cluster types were categorized into single adult female, single adult male, female group, male group, mixed group, Ewe-lamb group, and cub group.

The 7 cluster types are divided as follows:(1) Single adult female: only 1 female moving alone in the photo or video. (2) Single male group: only 1 male moving alone in the photo or video.(3) Female group: the presence of two or more female adult horse deer in the same activity area.(4) Male group: consists of two or more male red deer;(5) Mixed group: includes at least one adult female red deer and one adult male red deer; (6) Ewe-lamb group: females are often accompanied by juvenile or subadult individuals. (7) Cub group: contain only subadults or fawns.When recording, it is necessary to recognize the presence or absence of adult males in the cluster and to avoid recording mixed groups as cub groups.

  1. Based on the reserve's climatic characteristics, the year was divided into spring(April to May), summer (June to September), autumn (October to November), and winter (December to March).In this study, data were processed and analyzed using SPSS 26.0 and R4.2.1. Since the data were all non-normally distributed, seasonal differences in the frequency of occurrence of different cluster types were analyzed using R×C c2Test; seasonal differences in cluster size were analyzed using the Kruskal-Wallis H-test; and seasonal differences in cluster size of all deer and adult deer were analyzed using the paired t-test. A P-value of less than 0.05 represents a significant difference in the data. Sexual segregation and aggregation were tested using SASS.
  2. In discussing the results regarding the frequency and size of group types, we consider the links between sex segregation, group composition, and group size.In summary, the group size and group frequency of Tianshan wapiti are affected by the breeding cycle and season, and the birth of fawns increases the average group size of Tianshan wapiti by 1 to 2 animals in spring and summer. Summer coincides with the period of fawning in Tianshan wapiti when mixed groups decrease and ewe-lamb groups and female groups increase. As fall progressed, the average group size decreased and the frequency of mixed and single male herds increased. Tianshan wapiti are at low risk of predation, have sparse numbers of natural enemies such as wolves and lynx, and exhibit sex clustering in July and October through December.

Specific comments:

Line 2 and in the rest of the manuscript. I suggest to replace ‘group pattern’ by the more

commonly used ‘grouping patterns’.

We agree with the reviewers' suggestions and revised the manuscript as suggested.

Line 24. The verb ‘to adapt’ should be avoided here. No doubt that group size and composition vary through seasons in the study carried out by the authors. However, the authors provide no evidence that these variations correspond to adaptations set by natural selection, or even that they have positive effects on fitness (this is not the same thing; see the concept of ‘exaptation’ introduced by Gould & Vrba 1981). Nothing is simple in the case of grouping patterns. As suggested above, several aspects of grouping patterns are mechanically linked. In addition, it is likely that at least some of their variations are not adaptations but purely mechanical consequences, e.g. the increase of mean group size with (local) population density (Barrette 1991; Pepin & Gerard 2008).

Lines 28-29 ‘ultimately facilitating adaptability to their habitats’. See comment for line 24.

Lines 47-48 ‘These observed changes… adaptations to their environment’. See comment for line 24.

We agree with the reviewers' suggestions and revised the manuscript as suggested. See the concept of ‘exaptation’ introduced by Gould & Vrba in 1981. We have corrected the"adapt" into "exaptation".

Line 64. ‘environmental conditions’ is evasive. In relation to the cited references, I suggest

‘population density and distance at which individuals can perceive one another’.

We sincerely thank the reviewer for careful reading. As suggested by the reviewer, we have corrected the"environmental conditions" into "population density, and distance at which individuals can perceive one another".

Lines 78-81. I am not sure these sentences should be kept. If they are, a few references should be added.

We sincerely thank the reviewer for careful reading.We chose not to retain these sentences.

Lines 92-93. These sentences should be placed with the other sentences describing climate.

We sincerely thank the reviewer for careful reading.We have put these sentences together with other sentences describing the climate.

Line 94. ‘hm’ or km? ‘hm’ is odd: ‘hectare’ (ha) is more common.

We feel sorry for our carelessness. In our resubmitted manuscript, the typo is revised. Thanks for your correction.We have corrected the"hm2" into "km2".

Line 101 ‘984 days’. If there is no error, I don’t understand the sentence.

We feel sorry for our carelessness. In our resubmitted manuscript, the typo is revised. Thanks for your correction.We have corrected the"984" into "98.4".

Line 102. This sentence can be deleted.

We sincerely thank the reviewer for careful reading.This sentence has been deleted

Lines 115-120. Young under one year of age can likely be distinguished from the other age-sex classes.

We sincerely thank the reviewer for careful reading.We have distinguished between toddlers under one year of age and subadults.”While analyzing pictures from surveillance cameras, we distinguished red deer sex by the presence or absence of antlers and their age by body and antler sizes and shape of antlers and by the following characteristics: (1) Adult males: large-sized normally with branched antlers; when antlers cast in late Winter – early Spring, there are visible signs of shedding on the head. Sex is determined by body size and external sexual organs. (2) Adult females: medium-sized, antlerless, and often accompanied by calves. (3) Sub-adult individuals: Smaller individuals with slender and non-forked horns are male sub-adults.(4) Fawns: smaller, antlerless, usually following adult females; those born in the current year have a spotted pattern on both sides of the body.”

Lines 123-126. I suggest 6 group types: single adult female (with or without younger animals), female groups of at least 2 adults, mixed-sex groups, male groups of at least two adults, single males (with or without younger animals), and groups without adults. If the authors computed the group type frequencies that are expected under the hypothesis of no segregation (see main comment), the proportion p of females among the adults may refer to either all the adults observed during the season under focus, or only those observed in groups of at least 2 adults.

We agree with the reviewers' suggestions and revised the manuscript as suggested.The 7 group types are divided as follows:(1) Single adult female: only 1 female moving alone in the photo or video. (2) Single male group: only 1 male moving alone in the photo or video.(3) Female group: the presence of two or more female adult horse deer in the same activity area.(4) Male group: consists of two or more male red deer;(5) Mixed group: includes at least one adult female red deer and one adult male red deer; (6) Ewe-lamb group: females are often accompanied by juvenile or subadult individuals. (7) Cub group: contain only subadults or fawns.When recording, it is necessary to recognize the presence or absence of adult males in the cluster and to avoid recording mixed groups as cub groups.

Line 127. Meaning?

We sincerely thank the reviewer for careful reading.We chose not to retain these sentences.

Line 139. This way of quantifying group size was not introduced by Reiczigel et al (2008). As noted by these authors, Jarman already used it in Appendix 2 of his landmark paper published in 1974 on the social organisation of African antelope. After Jarman’s publication, this way of quantifying group size has often been referred to as ‘typical group size’ (see for instance Barrette 1991). Please improve this sentence.

We sincerely thank the reviewer for careful reading.We chose not to retain these sentences.

Lines 156-158. The authors rightly apply the Chi2 test to the frequencies of group types. As they surely know, it would have been erroneous to apply this test to the frequencies with which individuals were found in the different group types. As with crowding data, the problem lies in the non-independence of such individual-centred data.

We sincerely thank the reviewer for careful reading.We have re-selected our data analysis methods based on your suggestions.In this study, data were processed and analyzed using SPSS 26.0 and R4.2.1. Since the data were all non-normally distributed, seasonal differences in the frequency of occurrence of different cluster types were analyzed using R×C c2Test; seasonal differences in cluster size were analyzed using the Kruskal-Wallis H-test; and seasonal differences in cluster size of all deer and adult deer were analyzed using the paired t-test. A P-value of less than 0.05 represents a significant difference in the data. Sexual segregation and aggregation were tested using SASS.

Line 157. ‘activity’?

We sincerely thank the reviewer for careful reading.We have corrected the "activity" into "clustering frequency".

Line 165. ‘sub-adult females’? What about sub-adult males? What about the young under one year of age?

We sincerely thank the reviewer for careful reading.We have re-customized the definition of age groups with the following modifications:While analyzing pictures from surveillance cameras, we distinguished red deer sex by the presence or absence of antlers and their age by body and antler sizes and shape of antlers and by the following characteristics: (1) Adult males: large-sized normally with branched antlers; when antlers cast in late Winter – early Spring, there are visible signs of shedding on the head. Sex is determined by body size and external sexual organs. (2) Adult females: medium-sized, antlerless, and often accompanied by calves. (3) Sub-adult individuals: Smaller individuals with slender and non-forked horns are male sub-adults.(4) Fawns: smaller, antlerless, usually following adult females; those born in the current year have a spotted pattern on both sides of the body.

Between line 167 and section 3.1. In my opinion, the authors should insert here a figure showing the seasonal variations in the observed proportions of the different age-sex classes (or at least the observed proportions of females and males among adults). Of course, a paragraph commenting on the figure is needed.

Line 168 ‘32’. In the Simple summary and Abstract, maximum group size is 39.

We apologize for our carelessness. We have rechecked the data and it is “32”.This has been corrected in our resubmission of the manuscript. Thank you for your correction.

Figure 1. This figure and its caption can remain in the revised version of the manuscript, even if variations in group size are analysed using group size rather than crowding data.

We sincerely thank the reviewer for careful reading.We retained Figure 1.

Lines 183-186. Seasons should rather be defined in the section Materials and methods.

We sincerely thank the reviewer for careful reading.Seasons have been placed in the “Materials and Methods” section.

Line 185 ‘nursing period’. I suspect that hinds are still suckling calves in July. In this case, the name ‘birthing period’ would certainly be more appropriate than ‘nursing period’ for May-June. However, if births rarely occur in May (I suspect they mainly occur in June), a more appropriate name would be simply ‘spring’, as May-June undoubtedly corresponds to the beginning of the growing season.

Line 186. I suspect that most of the fecundations occur in October, in which case ‘Fall’ (American English) or ‘Autumn’ (British English) might be more appropriate for OctoberDecember.

We sincerely thank the reviewer for careful reading. We reclassified the seasons, and we know from the literature the breeding and rutting periods of the Tianshan wapiti, which helped us to analyze the grouping patterns.

Based on the reserve's climatic characteristics, the year was divided into spring (April to May), summer (June to September), autumn (October to November), and winter (December to March).Tianshan wapiti rut and mate in September-October, and give birth to fawns in May-July.

Line 186. The paper by Wang et al (2018) concerns the Siberian ibex. In addition, in Animals, references must be indicated by numbers in square brackets.

We sincerely thank the reviewer for careful reading.We will pay attention to the citation format of the references.

Line 188 or its neighbourhood. In my opinion, the authors should insert a figure showing the seasonal variations in group size (all group types combined) and a paragraph commenting on the figure. Actually, I think that the figure should include two curves: a curve showing the variation of mean group size ± SE when all the individuals are considered, and a similar curve when only adults are considered. This would show to which extent births are responsible of the fluctuations in group size.

We agree with the reviewers' suggestions and revised the manuscript as suggested.

Fig. 3 Seasonal variation in mean group change for all individuals (n = 2335) and mean group size for only adult individuals (n = 2130)

Line 189. What the authors call ‘average group size’ is actually the mean value of the crowding index. The authors, if they wish, can indicate the mean value of the crowding index (per season and/or group type) in the main text or a table. However, this mean value should not be accompanied by SD, SE or any other confidence interval computed using standard formulas since the latter assume data independence.

Lines 189-207. Give means (and standard errors, if any) with only two decimals.

We sincerely thank the reviewer for careful reading.We use the mean value(±standard error) to represent the size of the cluster. Also, only two decimal places are kept.

Lines 188-195. A question arises here. Female groups are larger than male groups. This may be partly due to the sex ratio (see main comment). However, this may also be due to the fact that an adult female is often accompanied by her calf. So, rather than comparing the total number of individuals, it would be more appropriate to compare the number of adults between male groups and female groups. In addition, mixed-sex groups include by definition at least two adults. So, it is the number of adults in groups of at least two adults that should be considered when comparing female or male groups with mixed-sex groups. Group size (all age-sex classes included) can be mentioned, but is finally less relevant.

Lines 188 and 195. Two degrees of freedom (df) are required for standard F test.

Lines 190 and 192. The df of residuals is required for standard t test.

Table 1 and Figure 2. Average growing index should not be accompanied by SD or SE

computed using standard formulas. See also comment for lines 189-207.

We agree with the reviewers' suggestions and revised the manuscript as suggested.

Fig. 3. If the authors compute the frequencies expected under the hypothesis of no segregation (see main comment, and comment for lines 123-126), I suggest that Fig. 3 includes two histograms: a cumulated histogram showing for each season the observed frequencies of the five group types not including adults (see comment for lines 123-126), and a similar histogram for the frequencies expected under the hypothesis of no segregation. The type of figure suggested is as follows.

We agree with the reviewers' suggestions and revised the manuscript as suggested.

Fig. 4 Seasonal variation in the frequency of occurrence of group types

Lines 240-278 (Discussion). The authors do not study the (possible) consequences of the variations they observe in grouping patterns on fitness. So, rather than referring to speculative ultimate explanations, I suggest they favour proximal explanations involving the reproductive cycle, vegetation phenology (and patchiness?), possible seasonal variations in sex ratio, and the mechanical links between sex ratio, group size and group composition (see main comment). Could it be that males and females concentrate on growing patches in May-June, hence the high frequency and size of mixed-sex groups at that time? If the authors find that adult males and females tend to form separate groups outside the rutting season, they should comment on this as well.

We agree with the reviewers' suggestions and revised the manuscript as suggested.

Line 240 ‘population size’. Population?

Line 245. ‘for survival’ should be deleted. It might be argued that within-group competition affects reproduction before survival.

Line 246. I suggest ‘Conversely, individuals in larger groups would experience reduced predation pressure’.

Lines 274-275. The authors should delet this hypothesis, too unlikely for deer specialists.

We agree with the reviewers' suggestions and revised the manuscript as suggested.

The conclusion has no link with the rest of the manuscript. It must be entirely rewritten.

We agree with the reviewers' suggestions and revised the manuscript as suggested.We have rewritten the conclusion along with the rest of the manuscript.

We are grateful for the invaluable feedback provided by the reviewers, and we look forward to the opportunity to improve our manuscript in response to their guidance.

Round 2

Reviewer 1 Report

Comments and Suggestions for Authors

In my first review I wrote “The manuscript requires profound changes. The taxonomy must be updated, the definitions of age groups and more in general the specific terminology must be re-formulated, the discussion of the results must be quite improved. The authors do not seem to have a complete handle on the topic. Important works on red deer and wapiti biology are ignored.” Unfortunately I can repeat the same text for the new version.

After many changes most of the basic problems remain. The additions and corrections were made hastily. Most of my critics and suggestions do not seem to be absorbed. Most of the section on taxonomy has not been reformulated, the abundant references suggested by me (Polziehn & Strobeck 1998, Kuwayama & Ozawa 2000, Randi et al. 2001, Ludt et al. 2004, Lorenzini et al. 2005, Heckeberg 2020, Mackiewicz et al. 2022; Lovari et al. 2018 for C. elaphus, Brook et al. 2017 for C. hanglu and Brook et al. 2018 for C. canadensis in the IUCN Red List) were ignored. Also the suggestion to refer to Clutton-Brock et al. 1982 (a fundamental text on red deer biology including social behavior) was ignored. The definitions of age and sex classes, despite my precise suggestions, are still not clear and sometimes ambiguous, with serious errors also for the terminology of different classes (fawn, lamb, cub instead of kid or calf; ewe instead of female; subadult instead of yearling male, horns instead of antlers etc). The only real improvement was the introduction of the subject “segregation”. Some sentences are incomprehensible or confused, some terms are used without knowing the actual meaning (for ex. hermaphrodite, probably used instead of dimorphic). The English is often poor.

In my first review I added some final questions to help better interpret the data: “Why the sexual spatial and social segregation is so limited in this wapiti population? …Why mixed groups are frequent also in winter, when the rutting season is finished?” I have not found a convincing explanation for these results. But a scientific study consists not only in a description of the results but must include a credible interpretation of them.

The final paragraph on literature should be profoundly re-edited, eliminating three repetitions, including new titles (for example in taxonomy) and making uniform the style of quotations (lowercase letters for some titles, italics for scientific names etc).

The current version of the manuscript remains still insufficient to be published.

I  add some notes on single points:

Line 18: grouping behavior

Lines 22-24: better “Herd and group frequency of Tianshan wapiti were affected…, with sex aggregations from July…”

Line 27: grouping behavior

Lines 29-30: The sentence is senseless. Please reformulate “ The social organization of the Tianshan wapiti is poorly known”

Line 38: “and showed a tendency to…”

Lines 46-47: actually the reverse: “not only across different species of ungulates but also within the same species”

59 (and 297): with hermaphroditic bodies???!! Hermaphrodites are organisms with reproductive organs of both sexes, like earthworms and snails! Do you mean dimorphic? Which anomalies?

64-71: All this section should be reformulated following my previous suggestions.

117: age classes instead of “adult/young”

120 (and 135, 186, 237, 273, 301, 320): mother-kid instead of ewe-lamb

120 (and 136, 186): kid group

[ewe is only the female of a sheep, lamb is only the kid of a female sheep, cub can be used only for kids of Carnivores]

125: “Sex is determined…” should be cancelled.

126: In red deer and wapiti subadults are only 2-4 year-old males with branched antlers but not fully grown. You probably wish to refer to the yearling males, defined as “smaller individuals with slender and unbranched antlers”

128: calves or kids (not fawns, a term used for small-sized and medium-sized Cervids, roe deer, white-tailed deer etc but not for wapiti or red deer)

129: actually the spots are visible only in the first 2-3 months of life!

136: A ‘cub group’ (actually kid group) cannot contain subadults. You probably can use another definition like “juvenile group” or “young group”

150: “which account for”

150-152: the numbers do not add up: if you put together adult males, adult females and subadults (25,74% + 65,48% + 8,78%), you make 100%. And kids? So, what do you really mean with the category of subadults? Something is wrong.

154, 156: wapiti and not red deer

157 better ”…those from three [to what?] were 12,48%...”

159: were groups

164: better “groups of 7 and 9 individuals (7,39% and 15,12% respectively)”

222-223: accounting for

234: “and again a decreasing…”

Fig. 5 Tianshan Wapiti [at the top]

Wapiti instead of wapati in the caption below

259, 261, 263: wapiti instead of red deer

269: natural predators      please cancel “we found that”

270: better “wolves were not recorded”

286-288: this is a meaningless sentence. Please clarify.

293-295: the explanation is not clear. Why they aggregate?

298: Due to??

311: males are never also minimally involved in feeding and caring! How can they feed the kids?

313: In the fall…into autumn.   ?? [fall is an Americanism for autumn]

314: inter-fall??

320: calving

322: better “mixed herds and single males”

323: better “low number of natural predators”

Paragraph on literature:

Title n 11 repeated as n 31

Title n 13 Bowyer R.T. (not R.T. Bowyer) repeated as n 33

Title n 17 repeated as n 34

Lines 366-367: title in lowercase letters

Scientific names in italics

Comments on the Quality of English Language

The English is poor. Sometimes it is difficult to understand what the authors want to mean. The specific terminology for a Cervid like wapiti is ofter wrong.

Author Response

Respond to Reviewer#1’ comments

Comments 1: In my first review I wrote “The manuscript requires profound changes. The taxonomy must be updated, the definitions of age groups and more in general the specific terminology must be re-formulated, the discussion of the results must be quite improved. The authors do not seem to have a complete handle on the topic. Important works on red deer and wapiti biology are ignored.” Unfortunately I can repeat the same text for the new version.After many changes most of the basic problems remain. The additions and corrections were made hastily. Most of my critics and suggestions do not seem to be absorbed. Most of the section on taxonomy has not been reformulated, the abundant references suggested by me (Polziehn & Strobeck 1998, Kuwayama & Ozawa 2000, Randi et al. 2001, Ludt et al. 2004, Lorenzini et al. 2005, Heckeberg 2020, Mackiewicz et al. 2022; Lovari et al. 2018 for C. elaphus, Brook et al. 2017 for C. hanglu and Brook et al. 2018 for C. canadensis in the IUCN Red List) were ignored. Also the suggestion to refer to Clutton-Brock et al. 1982 (a fundamental text on red deer biology including social behavior) was ignored. The definitions of age and sex classes, despite my precise suggestions, are still not clear and sometimes ambiguous, with serious errors also for the terminology of different classes (fawn, lamb, cub instead of kid or calf; ewe instead of female; subadult instead of yearling male, horns instead of antlers etc). The only real improvement was the introduction of the subject “segregation”. Some sentences are incomprehensible or confused, some terms are used without knowing the actual meaning (for ex. hermaphrodite, probably used instead of dimorphic). The English is often poor.

In my first review I added some final questions to help better interpret the data: “Why the sexual spatial and social segregation is so limited in this wapiti population? …Why mixed groups are frequent also in winter, when the rutting season is finished?” I have not found a convincing explanation for these results. But a scientific study consists not only in a description of the results but must include a credible interpretation of them.

The final paragraph on literature should be profoundly re-edited, eliminating three repetitions, including new titles (for example in taxonomy) and making uniform the style of quotations (lowercase letters for some titles, italics for scientific names etc).

Responses 1: We sincerely thank the reviewer for careful reading. As suggested by the reviewer, we have profoundly revised the manuscript, redefining the definition of age groups and the definition of group types. Important works on red deer have been added, references [14-24]. Added to the Discussion section: (1) Why do the sexes of Tianshan wapiti aggregate in July and October-December? (2) Why are mixed groups also frequent in winter? In addition to being related to the reproductive cycle, this may be related to food resources and quality. And the format of references was modified.

We also invited an native English speaker help us revising the whole manuscript, and now the English writing is better.

Notes on specific points:

Comments 2: Line 18: grouping behavior

Responses 2: We sincerely thank the reviewer for careful reading. As suggested by the reviewer, we have corrected it to “ grouping behavior”

Comments 3: Lines 22-24: better “Herd and group frequency of Tianshan wapiti were affected…, with sex aggregations from July…”

Responses 3: As suggested by the reviewer, we have corrected it to “Group size and frequency of Tianshan wapiti were affected by reproductive cycles and seasons, with sex aggregations in July and from October to December. ”

Comments 4: Line 27: grouping behavior

Responses 4: As suggested by the reviewer, we have corrected it to “grouping behavior ”

Comments 5: Lines 29-30: The sentence is senseless. Please reformulate “ The social organization of the Tianshan wapiti is poorly known”

Responses 5: As suggested by the reviewer, we have corrected it to “The social organization of the subspecies of wapiti inhabiting the Tian Shan Mountains is poorly known”

Comments 6: Line 38: “and showed a tendency to…”

Responses 6: We sincerely thank the reviewer for careful reading. As suggested by the reviewer, we have corrected it to “ The SSAS results showed no segregation of female and male Tianshan wapiti in any month and a tendency to sex aggregation in July and from October to December.”

Comments 7: Lines 46-47: actually the reverse: “not only across different species of ungulates but also within the same species”

Responses 7: We sincerely thank the reviewer for careful reading. As suggested by the reviewer, we have corrected it to “ Variability exists not only across different species of ungulates, where group characteristics can differ significantly, but also within the same species.”

Comments 8: 59 (and 297): with hermaphroditic bodies???!! Hermaphrodites are organisms with reproductive organs of both sexes, like earthworms and snails! Do you mean dimorphic? Which anomalies?

Responses 8: We have re-written this part according to the Reviewer's suggestion.”In many mammals,males and females live solitarily or in separate groups outside the breeding season [11]. Sexual segregation is especially widespread in ungulates [12], although its causes remain debated. They may actually depend on species and involve sex-related differences in food preferences, habitat selection, energy expenditure, or activity rhythm [13].“

Comments 9: 64-71: All this section should be reformulated following my previous suggestions.

Responses 9: We have re-written this part according to the Reviewer's suggestion.

Comments 10: 117: age classes instead of “adult/young”

Responses 10: We sincerely thank the reviewer for careful reading. As suggested by the reviewer, we have corrected it to “ In contrast, individual wapiti were identified based on their coat color characteristics, body size, sex, and age classes. ”

Comments 11: 120 (and 135, 186, 237, 273, 301, 320): mother-kid instead of ewe-lamb

Responses 11: We sincerely thank the reviewer for careful reading. As suggested by the reviewer, we have  modified.

Comments 12: 120 (and 136, 186): kid group

Responses 12: We sincerely thank the reviewer for careful reading. As suggested by the reviewer, we have modified.

Comments 13: 125: “Sex is determined…” should be cancelled.

Responses 13: We sincerely thank the reviewer for careful reading.This sentence has been deleted

Comments 14: 126: In red deer and wapiti subadults are only 2-4 year-old males with branched antlers but not fully grown. You probably wish to refer to the yearling males, defined as “smaller individuals with slender and unbranched antlers”

128: calves or kids (not fawns, a term used for small-sized and medium-sized Cervids, roe deer, white-tailed deer etc but not for wapiti or red deer)

136: A ‘cub group’ (actually kid group) cannot contain subadults. You probably can use another definition like “juvenile group” or “young group”

Responses 14: We sincerely thank the reviewer for careful reading. As suggested by the reviewer, we have corrected it to “ While analyzing pictures from surveillance cameras, we distinguished wapiti sex and age using the following criteria.

(1)Adult males : large-sized normally with branched antlers ; when antlers cast in late Winter-early Spring, there are visible signs of shedding on the head.

(2)Adult females : medium-sized, antlerless and often accompanied by calves.

(3)Yearlings : smaller than adults and, if males, provided with non-branched and slender antlers.

Seven group types were distinguished:(1) Single female groups: only one adult female. (2) Single male groups: only one adult male. (3) Female groups: groups consisting of two or more adult females (4) Male groups: groups consisting of two or more adult males; (5) Mixed groups: groups consisting of females and males; (6) Mother-kid groups: groups consisting of females and kids, excluding males (7) Young groups: containing subadults or calves.“

Comments 15: 150: “which account for”

150-152: the numbers do not add up: if you put together adult males, adult females and subadults (25,74% + 65,48% + 8,78%), you make 100%. And kids? So, what do you really mean with the category of subadults? Something is wrong.

Responses 15: We sincerely thank the reviewer for careful reading. As suggested by the reviewer, we have corrected it to“Throughout the study period, we observed a total of 609 groups, whose 2335 members were 601 adult males (i.e., 25.74% of all individuals), 1529 adult females (65.48%), and 205 young individuals (8.78%).”

Comments 16: 154, 156: wapiti and not red deer

Responses 16: We sincerely thank the reviewer for careful reading. As suggested by the reviewer, we have  modified.

Comments 17: 157 better ”…those from three [to what?] were 12,48%...”

159: were groups

164: better “groups of 7 and 9 individuals (7,39% and 15,12% respectively)”

Responses 17: We sincerely thank the reviewer for careful reading. As suggested by the reviewer, we have  modified.

Comments 18: 222-223: accounting for

234: “and again a decreasing…”

Fig. 5 Tianshan Wapiti [at the top]

Wapiti instead of wapati in the caption below

Responses 18: We sincerely thank the reviewer for careful reading. As suggested by the reviewer, we have  modified.

Comments 19: 259, 261, 263: wapiti instead of red deer

269: natural predators please cancel “we found that”

270: better “wolves were not recorded”

Responses 19: We sincerely thank the reviewer for careful reading. As suggested by the reviewer, we have  modified.

Comments 20: 293-295: the explanation is not clear. Why they aggregate?

311: males are never also minimally involved in feeding and caring! How can they feed the kids?

Responses 20: We sincerely thank the reviewer for careful reading. As suggested by the reviewer, we have added to the Discussion section: (1) Why do the sexes of Tianshan wapiti aggregate in July and October-December? (2) Why are mixed groups also frequent in winter? In addition to being related to the reproductive cycle, this may be related to food resources and quality. Details can be found in the revised manuscript.

Comments 21: 313: In the fall…into autumn.   ?? [fall is an Americanism for autumn]

314: inter-fall??

320: calving

322: better “mixed herds and single males”

323: better “low number of natural predators”

Responses 21: We sincerely thank the reviewer for careful reading. As suggested by the reviewer, we have  modified.

Comments 22: Paragraph on literature:

Title n 11 repeated as n 31

Title n 13 Bowyer R.T. (not R.T. Bowyer) repeated as n 33

Title n 17 repeated as n 34

Lines 366-367: title in lowercase letters

Scientific names in italics

Responses 22: We sincerely thank the reviewer for careful reading. As suggested by the reviewer, we have  modified.

Reviewer 2 Report

Comments and Suggestions for Authors

Author Response

Respond to Reviewer#2’ comments

Comments 1: The authors of the manuscript have made most of the changes I proposed in my first report, for which I am grateful. As detailed below, I think that a number of minor revisions should still be made before the manuscript is published in Animals. I hope that these final proposals will be helpful.

Responses 1: We sincerely thank the reviewer for their valuable feedback, which we have used to improve the quality of our manuscript.

Comments 2: Line 18. According to genetics, the Tianshan subspecies is related to the East Asian and North American subspecies of red deer. As a consequence, ‘wapiti’ is correct. However, the authors must be consistent about the scientific names they use. On line 68, all red deer subspecies are implicitly presented as subspecies of Cervus elaphus, whereas on line 74 and in the title of the manuscript, the Tianshan deer is implicitly considered to be a subspecies ofCervus canadensis. The authors can use ‘wapiti’ as a common name for Tianshan deer, but they must choose between a single species of red deer (Cervus elaphus) or several species(Cervus elaphus, C. canadensis, C. hanglu). See also comments for lines 68-72 and line 74.

Responses 2: We agree with the reviewers' suggestions and wilincorporate the recommended changes into the manuscript.

“In this study, we tested the pattern of groups of Cervus canadensis songaricus [14-24] living in Tianchi Bogda Peak, Xinjiang, China. The Tianshan wapiti is mainly distributed in the alpine zone of mixed coniferous and broad-leaved forests, forest shrubs, and grasslands on the northern slopes of the Tianshan Mountains at elevations of 1800 to 3200 meters. In this study, we will analyze the seasonal variation in the group size of Tianshan wapiti, as seasonality affects the group. In addition, we tested whether the groups observed throughout the annual cycle could be the result of sex-independent associations, applying the ‘sexual segregation and aggregation statistic'(SSAS) to the numbers of adult males and females in the groups.”

Comments 3: Lines 18-19. In suggest ‘Grouping patterns of the wapiti subspecies living in…’ and ‘, were investigated…’.

Responses 3: We sincerely thank the reviewer for careful reading. As suggested by the reviewer, we have corrected it to“Grouping patterns of the wapiti subspecies living in the Bogda Peak Nature Reserve, Tianshan Mountain, Northwest China, were investigated from July 2019 to November 2020 using infrared-triggered camera-trapping.”

Comments 4: Line 20. Please replace 39 by 32 (as indicated in the Results section).

Responses 4: We feel sorry for our carelessness. In our resubmitted manuscript, the typo is revised.We have corrected the"39" into "32".

Comments 5: Line 23 and all the other occurrences of ‘SASS’ in the manuscript: replace ‘SASS’ by ‘SSAS’.

Responses 5: We sincerely thank the reviewer for careful reading. As suggested by the reviewer, we have  modified.

Comments 6: Line 24. I suggest ‘in July and from October to December’.

Responses 6: We sincerely thank the reviewer for careful reading. As suggested by the reviewer, we have corrected it to“Group size and frequency of Tianshan wapiti were affected by reproductive cycles and seasons, with sex aggregations in July and from October to December.”

Comments 7: Line 27. I suggest ‘Grouping behavior’.

Responses 7: We sincerely thank the reviewer for careful reading. As suggested by the reviewer, we have  modified.

Comments 8: Lines 29-30. I suggest ‘The social organization of the subspecies of wapiti inhabiting the Tian Shan Mountains is poorly known’.

Responses 8: We sincerely thank the reviewer for careful reading. As suggested by the reviewer, we have  modified.

Comments 9: Line 32. ‘-type’ can be deleted.

Responses 9: We sincerely thank the reviewer for careful reading.The ‘-type’ has been deleted

Comments 10: Line 34. Please replace 39 by 32.

Responses 10: We feel sorry for our carelessness. In our resubmitted manuscript, the typo is revised.We have corrected the"39" into "32".

Comments 11: Line 57. I suggest ‘Moreover, in many mammals, …’.

Line 58-63. ‘hermaphroditic bodies’? I suggest rewriting these lines as follows: ‘Sexual segregation is especially widespread in ungulates [12], although its causes remain debated. They may actually depend on species and involve sex-related differences in food preferences, habitat selection, energy expenditure, or activity rhythm [13].’

Responses 11: We sincerely thank the reviewer for careful reading. As suggested by the reviewer, we have corrected it to“Moreover, in many mammals,males and females live solitarily or in separate groups outside the breeding season [11]. Sexual segregation is especially widespread in ungulates [12], although its causes remain debated. They may actually depend on species and involve sex-related differences in food preferences, habitat selection, energy expenditure, or activity rhythm [13].”

Comments 12: Line 66. According to phylogenies based upon DNA, ‘Artiodactyla’ should be replaced by ‘Cetartiodactyla’.

Lines 68-72. If the authors choose to say that there are several species of red deer,

they must rewrite these sentences. Alternatively, if they choose to say that there is a single species, they must change the scientific names they use line 74 (see below).

Line 74. I suggest either ‘Altai wapiti (Cervus e. sibiricus), Tianshan wapiti (C. e. songaricus), and Tarim hangul (C. e. yarkandensis)’ or ‘Altai wapiti (Cervus canadensis sibiricus), Tianshan wapiti (C. c. songaricus), and Tarim hangul (C. hanglu yarkandensis)’. According to recent analyses of mitochondrial DNA, the hangul is not a wapiti.

Responses 12: We sincerely thank the reviewer for careful reading. As suggested by the reviewer, we have corrected it to“In this study, we tested the pattern of groups of Cervus canadensis songaricus [14-24] living in Tianchi Bogda Peak, Xinjiang, China. The Tianshan wapiti is mainly distributed in the alpine zone of mixed coniferous and broad-leaved forests, forest shrubs, and grasslands on the northern slopes of the Tianshan Mountains at elevations of 1800 to 3200 meters. In this study, we will analyze the seasonal variation in the group size of Tianshan wapiti, as seasonality affects the group. In addition, we tested whether the groups observed throughout the annual cycle could be the result of sex-independent associations, applying the ‘sexual segregation and aggregation statistic'(SSAS) to the numbers of adult males and females in the groups.”

Comments 13: Lines 85-86. I suggest to delete the sentence ‘The temperature varies greatly during a day.’

Responses 3: We sincerely thank the reviewer for careful reading.We chose not to retain these sentences.

Comments 14: Lines 117-120. In my opinion, these sentences can be deleted. Little information is given beyond what is detailed in the following two paragraphs.

Responses 14: We sincerely thank the reviewer for careful reading.We chose not to retain these sentences.

Comments 15: Lines 121-123. I suggest ‘…, we distinguished wapiti sex and age using the following criteria.

Responses 15: We sincerely thank the reviewer for careful reading. As suggested by the reviewer, we have corrected it to“While analyzing pictures from surveillance cameras, we distinguished wapiti sex and age using the following criteria.

(1)Adult males : large-sized normally with branched antlers ; when antlers cast in late Winter-early Spring, there are visible signs of shedding on the head.

(2)Adult females : medium-sized, antlerless and often accompanied by calves.

(3)Yearlings : smaller than adults and, if males, provided with non-branched and slender antlers.”

Comments 16: Line 125. The sentence ‘Sex is determined by body size and external sexual organs.’ should be deleted.

Responses 16: We sincerely thank the reviewer for careful reading.We chose not to retain these sentences.

Comments 17: Lines 126-128 and the rest of the manuscript. Use either ‘calf’ or ‘fawn’, but avoid using both.

Responses 17: We sincerely thank the reviewer for careful reading. As suggested by the reviewer, we have  modified.

Comments 18: Line 130. I suggest: ‘Seven group types were distinguished:’. Please, do not use ‘cluster’ as an alternative to ‘group’.

Line 132. ‘female horse deer in the same activity area’ —> ‘adult females’.

Line 135. ‘Ewe-lamb’ is not appropriate. Use ‘Female-young’ or ‘Female-fawn’ or ‘Femalecalf’ (see comment for lines 126-128).

Line 136. ‘Cub’ —> ‘Young’, ‘Calf’ or ‘Fawn’. See comment for lines 126-128.

Responses 18: We sincerely thank the reviewer for careful reading. As suggested by the reviewer, we have  modified the manuscript as suggested.

Comments 19: Lines 136-138. The authors should delete this last sentence.

Responses 19: We sincerely thank the reviewer for careful reading.We chose not to retain these sentences.

Comments 20: Lines 141-147. I suggest ‘As group size distribution was far from normal, we tested seasonal differences in mean group size using Kruskal-Wallis H-test. Seasonal variations in the observation frequencies of the different group types were tested using R×C 2-test. In addition, we tested whether the groups observed throughout the annual cycle could be the result of sex-independent associations, applying the ‘sexual segregation and aggregation statistic’ (SSAS) to the numbers of adult males and females in the groups.[17]. All statistical analyses were performed using SPSS 26.0 and R 4.2.1.’ Let me outline here that ‘RxC c2 test’ is odd, and that the paired t-test performed by the authors is irrelevant (see comment for lines 209-211).

Responses 20: We sincerely thank the reviewer for careful reading. As suggested by the reviewer, we have  modified.”As group size distribution was far from normal, we tested seasonal differences in mean group size using the Kruskal-Wallis H-test. Seasonal variations in the observation frequencies of the different group types were tested using R×C c2-test. In addition, we tested whether the groups observed throughout the annual cycle could be the result of sex-independent associations, applying the ‘sexual segregation and aggregation statistic’(SSAS) to the numbers of adult males and females in the groups[28].All statistical analyses were performed using SPSS 26.0 and R 4.2.1.”

Comments 21: Lines 143-145. Please, do not use ‘cluster’ as an alternative to ‘group’.

Line 150. Delete ‘which’.

Responses 21: We sincerely thank the reviewer for careful reading. As suggested by the reviewer, we have  modified.

Comments 22: Lines 149-152. I suggest: ‘Throughout the study period, we observed a total of 609 groups, whose 2335 members were 601 adult males (i.e. 25.74% of all individuals), 1529 adult females (65.48%) and 205 sub-adults (8.78%).’ However, if the sub-adults were only males, this should be stated explicitly. In addition, reading these lines, I understand that fawns(/calves) are not included in the 2335 individuals, which is probably wrong. Please improve the proposed sentence.

Responses 22: We sincerely thank the reviewer for careful reading. As suggested by the reviewer, we have  modified.”Throughout the study period, we observed a total of 609 groups, whose 2335 members were 601 adult males (i.e., 25.74% of all individuals), 1529 adult females (65.48%), and 205 young individuals (8.78%). ”

Comments 23: Line 168. ‘of different group types” should be deleted.

Line 170. The first sentence should be deleted.

Line 173. H value cannot be negative.

Line 186. Do not use ‘ewe-lamb’. See comment for line 135.

Line 187. Do not use ‘cub’. See comment for line 136.

Responses 23: We sincerely thank the reviewer for careful reading.We chose not to retain these sentences.

Comments 24: Figs 2 and 3. I suggest reversing Figs 2 and 3 (and their numbers). See also following

comment and comment for lines 209-211.

Lines 170-175. I suggest: ‘Kruskal-Wallis H-test revealed significant seasonal differences in group size (H = 28.91, df = 3, P < 0.0001) with a maximum in spring (Fig. 2; spring vs any other season: H-test, df = 1, P < …). Similar results were obtained when group size was calculated ignoring fawns [or calves], showing that the spring maximum was not due to newborns (Fig. 2; four seasons: H = …, df = 3, P < 0.0001; spring vs any other season: P < …).

Responses 24: We sincerely thank the reviewer for careful reading. As suggested by the reviewer,we have inverted Figures 2 and 3 (and their figures).”Kruskal-Wallis H-test revealed significant seasonal differences in group size (H = 28.91, df = 3, P < 0.0001) with a maximum in spring (Fig. 2; spring vs any other season: Spring VS Summer : H = 19.69,df = 1,P < 0.01;Spring VS Autumn : H = 24.25,df = 1,P <0.01;Spring VS Winter : H = 18.525,df = 1,P < 0.01;). Similar results were obtained when group size was calculated ignoring calves, showing that the spring maximum was not due to newborns (Fig. 2; four seasons: H = 22.59, df = 3, P < 0.0001; spring vs any other season: Spring VS Summer : H = 17.85,df = 1,P < 0.01;Spring VS Autumn : H =19.13,df = 1,P < 0.01;Spring VS Winter : H = 13.267,df = 1,P < 0.01).”

Comments 25: Lines 194-201. I suggest ‘Furthermore, we also found seasonal differences in group size for mixed groups (H = 12.40, df = 3, P < 0.01) and female-young groups (H = 8.32, df = 2, P < 0.02). Size of mixed groups was significantly larger in spring than in summer (H = 31.99, df = 1, P < 0.0001) and autumn (H = 40.00, df = 1, P < 0.0001), and size of female-young groups were much larger in spring than summer (H = 23.24, df = 1, P < 0.0001; Table 1; Fig. 3).’Fig. 2 (—> Fig. 3 if the authors follow my proposal) is lacking.Lines 209-211. These lines should be deleted: no test is needed to show that groups have a smaller size when part of the individuals are ignored (let me add that a test that compares two paired-samples of four values cannot be significant: sample size is too small). The only point in calculating mean group size without young is to check that spring maximum is not due to newborns. See comment for lines 170-175.

Responses 25: We agree with the reviewers' suggestions and wilincorporate the recommended changes into the manuscript.”Furthermore, we also found seasonal differences in group size for mixed groups (H = 12.40, df = 3, P < 0.01) and mother-kid groups (H = 8.32, df = 2, P < 0.02). Size of mixed groups was significantly larger in spring than in summer (H = 31.99, df = 1, P < 0.0001) and autumn (H = 40.00, df = 1, P < 0.0001), and size of mother-kid groups were much larger in spring than summer (H = 23.24, df = 1, P < 0.0001; Table 1; Fig. 3)”

Comments 26: Line 218. ‘change’ should be deleted.

Responses 26: We sincerely thank the reviewer for careful reading. As suggested by the reviewer, we have  modified.

Comments 27: Lines 221-227. I suggest ‘The 609 groups observed throughout the study period were 91 single adult females (i.e. 14.94% of all groups), 92 single adult males (15.11%), …, and seven young groups (1.15%). Nonetheless, group type frequencies varied according to season (�! = 93.32, df = 18, P < 0.0001; Fig. 4).’

Responses 27: We agree with the reviewers' suggestions and wilincorporate the recommended changes into the manuscript.”The 609 groups observed throughout the study period were 91 single adult females (i.e. 14.94% of all groups), 92 single adult males (15.11%), 121 female groups (19.87%), 54 male groups (8.87%), 196 mixed groups (32.18%), 48 mother-kid groups (7.89%) and seven young groups (1.15%). Nonetheless, group type frequencies varied according to season (2 = 93.32, df = 18, P < 0.05; Fig. 4)”

Comments 28: Lines 228-238. Relative frequencies are not independent: their sum is equal to 100%. So, instead of the current paragraph, I suggest a paragraph describing results season by season (the authors do this in the Discussion, but for lines 299-304, I suggest alternative sentences integrating group size and frequencies). My proposal for lines

228-238 is as follows. ‘In spring, mixed groups were the most frequent group type, while single adult females, single adult males, female groups and male groups exhibited their lowest frequencies of the year. In summer, frequency of mixed groups fell to its lowest value, while that of all other group types increased. In autumn, the frequency of single adult males reached its highestvalue while that of male groups was back to its lowest value; simultaneously, the frequencyof mixed groups re-increased, and female-young groups were no longer observed. In winter, frequencies of single adult males and mixed groups decreased, while those of the other group types re-increased.’

Responses 28: We sincerely thank the reviewer for careful reading. As suggested by the reviewer, we have  modified.

Comments 29: Line 245. ‘exhibited’ —> ‘found’

Line 252. ‘population size’ —> ‘group size’

Line 258. ‘larger groups experience’ —> ‘individuals in larger groups are assumed to

experience’

Responses 29: We sincerely thank the reviewer for careful reading. As suggested by the reviewer, we have  modified.

Comments 301: Lines 268-271. I suggest ‘In addition, according to our camera-trap survey of the birds and mammals of the Bogda Peek study area [15], we found that the Tianshan wapiti has very few predators since the wolf (Canis lupus) was never observed, and the Eurasian lynx only rarely. Therefore…’

Line 272. ‘by clustering’ —> ‘by forming in large groups’

Responses 30: We sincerely thank the reviewer for careful reading. As suggested by the reviewer, we have  modified.

Comments 31: Lines 273-275. This sentence is unclear.

Responses 31: We sincerely thank the reviewer for careful reading.We chose not to retain these sentences.

Comments 32: Lines 273-275. This sentence is unclear.

Lines 276-281. I do not agree with this reasoning. In spring, female-young groups are rarer than female groups, but they are much larger (Table 1). In addition, according to your definition of group types, mixed-sex groups can include adult females with young.

Responses 32: We sincerely thank the reviewer for careful reading. As suggested by the reviewer, we have  modified.

Comments 33: Lines 281-285. Theses sentences should be deleted.

Responses 33: We sincerely thank the reviewer for careful reading.We chose not to retain these sentences.

Comments 34: Lines 286-288. I suggest ‘As Tianshan wapiti exhibits strong dimorphism in body size, adult males and females could be expected to form mainly single-sex groups [19-22]. In contrast, we found that there was no sexual segregation at any time of the year, and that the two sexes tended to aggregate in July and from October to December.’

Responses 34: We sincerely thank the reviewer for careful reading. As suggested by the reviewer, we have  modified.

Comments 35: Line 289. ‘clustering’ —> ‘aggregation’

Line 291. ‘red deer’ —> ‘European red deer’

Line 292. ‘April to July’ —> ‘April to June’

Lines 293-296. I suspect that in Bonenfant’s study area, red deer do not gather in large herds in April-May as they do in the Tianshan Mountains, and that this explains the difference in spring.

Lines 299-305. I suggest: ‘According to our results, Tianshan wapitis gather in larger herds in spring than at any other time of the year. Correlatively, spring groupings very often include adults of both sexes. In the same season, female-young groups are much rarer than mixed herds, but exceptionally large with the birth of fawns. In summer, groups become much smaller than in spring. Correlatively, mixed groups become much rarer, while all the group types (solitary adults included) become more common than in the previous season. In autumn, group size remains limited. However, with the rut, male groups are rare, single adult males are frequent, mixed groups re-increase in frequency, and female groups are no longer observed. In winter, after the rutting season, mean group size increases only slightly, but all group types are re-observed, with a low frequency for single adult males.’

Responses 35: We sincerely thank the reviewer for careful reading. As suggested by the reviewer, we have  modified.

Comments 36: Line 306. Avoid quoting statistical results in the Discussion.

Lines 307-308. All births do not occur in June: a number of them clearly occur in May. I suspect that the decrease in mean group size between spring and summer is primarily due to a change in forage distribution and to the deer’s food requirement after winter.

Responses 36: We sincerely thank the reviewer for careful reading. As suggested by the reviewer, we have profoundly revised the manuscript, redefining the definition of age groups and the definition of group types. Important works on red deer have been added, references [14-24]. Added to the Discussion section: (1) Why do the sexes of Tianshan wapiti aggregate in July and October-December? (2) Why are mixed groups also frequent in winter? In addition to being related to the reproductive cycle, this may be related to food resources and quality. And the format of references was modified.

We are grateful for the invaluable feedback provided by the reviewers.. The authors hope these explanations would answer your doubts.